# Expression and Regulation of a Novel Decidual Cells-Derived Estrogen Target during Decidualization

**DOI:** 10.3390/ijms24010302

**Published:** 2022-12-24

**Authors:** Lin Lu, Yingni Chen, Zhenshan Yang, Shijin Liang, Songqi Zhu, Xiaohuan Liang

**Affiliations:** College of Veterinary Medicine, South China Agricultural University, Guangzhou 510642, China

**Keywords:** decidualization, embryo implantation, uterus, estrogen, *Cstb*

## Abstract

During decidualization in rodents, uterine stromal cells undergo extensive reprogramming to differentiate into distinct cell types, forming primary decidual zones (PDZs), secondary decidual zones (SDZs), and layers of undifferentiated stromal cells. The formation of secondary decidual zones is accompanied by extensive angiogenesis. During early pregnancy, besides ovarian estrogen, de novo synthesis of estrogen in the uterus is essential for the progress of decidualization. However, the molecular mechanisms are not fully understood. Studies have shown that Cystatin B (*Cstb*) is highly expressed in the decidual tissue of the uterus, but the regulation and mechanism of *Cstb* in the process of decidualization have not been reported. Our results showed that *Cstb* was highly expressed in mouse decidua and artificially induced deciduoma via in situ hybridization and immunofluorescence. Estrogen stimulates the expression of *Cstb* through the Estrogen receptor (ER)α. Moreover, in situ synthesis of estrogen in the uterus during decidualization regulates the expression of *Cstb*. Silencing the expression of *Cstb* affects the migration ability of stromal cells. Knockdown *Cstb* by siRNA significantly inhibits the expression of *Dtprp*, a marker for mouse decidualization. Our study identifies a novel estrogen target, *Cstb*, during decidualization and reveals that *Cstb* may play a pivotal role in angiogenesis during mouse decidualization via the *Angptl7*.

## 1. Introduction

Embryo implantation and decidualization are multi-step processes. After the embryo invasion, the trophoblast cells pass through the basement membrane of the luminal epithelial cells, build a connection with the stromal cells, and then the decidualization begins [1]. During the reproductive cycle, the decidualization of endometrial stromal cells lays the foundation for implantation and placentation. Steroid hormones, like estrogen and progesterone, support the process of decidualization and the early events of endometrial formation occurring at the maternal–fetal interface, playing an important role in the success of pregnancy and fetal development [2]. Estrogen receptor (ER) α is the main form of ER in the mouse uterus [3]. Studies have confirmed that injection of the estrogen antagonist ICI182780 into mice can severely block the expression of decidualization-related genes, indicating that estrogen can regulate this process through ER [4]. In addition, P450 aromatase can convert testosterone into estrogen [5], and injection of letrozole, an aromatase inhibitor, into pregnant mice can significantly inhibit the process of decidualization [6]. All the above studies indicate that estrogen is synthesized in the uterus and maintains the decidual process.

Decidualization is a gradual process that begins around the terminal spiral arteries in the superficial layers of the endometrium and, eventually, involves the entire endometrium. During the initial stages of embryo implantation, the metabolic exchange between mother and embryo is entirely decided by the spread on the thin layer of avascular decidual cells that surround the embryo, an area more clearly demarcated in the mouse uterus, known as the “primary decidua zone” (PDZ) [7]. With the development of decidualization, the stromal cells surrounding the primary decidual zone, are densely vascularized, and then develop into the “secondary decidual zone” (SDZ). During early pregnancy, the vascular system functions to enhance embryonic growth and development [8,9]. Progesterone regulates vascular elongation, subepithelial capillary plexus maturation, and growth and coiling of spiral arterioles [10,11]. Vascular endothelial growth factor (VEGF) family members and their receptors are critical for progesterone-induced endometrial vascular remodeling [12].

Cystatin B (*Cstb*) is widely expressed in most tissues and cell types and is believed to inhibit the cathepsin protease family by neutralizing the proteases leaking from the lysosome [13]. It has been reported in the literature that the cystatin family can change the structural, physical, and biochemical, properties of the extracellular matrix (ECM) [14] and may contribute to the invasion of cancer cells and metastasis by regulating ECM degradation [15]. In addition, new evidence proves that inhibiting *Cstb* in tumor cells can reduce tumor metastasis in vivo [16], indicating that *Cstb* may affect cell migration. Some studies have reported that *Cstb* was present in the uterine cavity fluid of early gestation cows [17]. It was also found that *Cstb* is highly expressed in female uterine decidual tissue, but its specific mechanism has not been studied in detail [18].

In this study, we found that *Cstb* is strongly expressed in mouse decidua. Silencing the expression of *Cstb* affects the migration ability and decidualization of stromal cells. Furthermore, *Cstb* may regulate angiogenesis during mouse decidualization via the *Angptl7*. Our study shows, for the first time, that *Cstb* acts as a *Angptl7* mediator during the progression of decidualization in mouse early pregnancy.

## 2. Results

### 2.1. Cstb Expression in Mouse Uterus during Early Pregnancy

In situ hybridization was used to localize the expression of *Cstb* mRNA in mice uteri from days 1 to 8 of pregnancy. From days 1 to 4, no expression of *Cstb* mRNA could be observed in the mice uteri. On day 5, the signal for *Cstb* mRNA was first detected in the subluminal stroma surrounding the implanting blastocyst at implantation sites (Figure 1A). Then, from days 6 to 8, an increase in expression of *Cstb* mRNA could be found in the decidual cells surrounding the embryos (Figure 1A). RT-qPCR was used to quantify the expression levels of *Cstb* mRNA in mice uteri from day 5 to day 8 of pregnancy. The result showed that the expression level in the mice uteri kept increasing through days 5 to 8 (Figure 1B). Immunofluorescence data also provided similar results. On day 5, the signal could be seen in the stromal cells surrounding the blastocyst. From days 6 to 8, the *Cstb* signal was detected in the decidual area around the blastocyst (Figure 1C).

### 2.2. Regulatory Role of Estrogen on Cstb Expression

To study the regulation of hormones on *Cstb* expression, ovariectomized mice were injected with 17β-estradiol (100 ng per mouse) and progesterone (1 mg per mouse), respectively. The RT-qPCR results showed that estrogen could contribute to the induction of expression of the *Cstb*, whereas progesterone failed to promote this effect (Figure 2A). To explore the effects of estrogen on *Cstb* in a short time, we tested whether the induction of *Cstb* was strengthened by 3 h and 24 h of 17β-estradiol treatment. The study showed that the expression of *Cstb* was significantly increased after 17β-estradiol treatment for 3 h and, subsequently, decreased (Figure 2B). Therefore, cells, after 3 h of 17β-estradiol treatment, were detected. In addition, after pretreatment with the ER antagonist ICI182780, the promoting effect of 17β-estradiol injections on the expression of *Cstb* mRNA abated (Figure 2C), indicating that its regulation of *Cstb* was ER-mediated.

Since ERα is the main receptor of estrogen [19], we tested whether ERα was involved in the regulation of estrogen on *Cstb*. Propyl pyrazole triol (PPT) is an ERα selective agonist and significant up-regulator of *Cstb* in the uterus of ovariectomized mice (Figure 2D). To further confirm the role of ERα, we evaluated the effect of estrogen regulation on *Cstb* expression in ovariectomized ERα knockout (ERαKO) mice and wild type (WT) littermates. The results showed that there was no significant change in *Cstb* expression in the ERαKO group, but the expression of *Cstb* was significantly induced by 17β-estradiol in WT littermates (Figure 2E). These data demonstrated that ERα mediated the estrogen-induced *Cstb* expression.

### 2.3. The Function of Cstb during Decidualization

In the mouse uterus, stromal cells undergo decidualization after embryo attachment [2]. In addition, it is also feasible to artificially induce decidualization by injecting oil into the pseudopregnant uterus. The result of doing so showed that by injecting in pseudopregnant mice on the morning of day 4, a strong *Cstb* mRNA signal could be observed in the subluminal stromal cells on day 8 (Figure 3A). After artificially induced decidualization, the expression of *Cstb* showed a marked up-regulation (Figure 3B). Immunohistochemistry further demonstrated the *Cstb* protein expression in the decidual cells in the pseudopregnant mice on day 8 (Figure 3C). Inhibition of aromatase-involved endogenous estrogen production significantly blocked decidualization [6]. To verify whether endogenous estrogen in the uterus induced the expression of *Cstb*, we injected letrozole into artificially decidualized mice to prevent the production of estrogen. As a result, the expression of *Cstb* in the decidua+letrozole group was significantly lower than in the normal decidua group (Figure 3D,E). Our data indicated that ERα mediated the regulation of endogenous estrogen on *Cstb* induction.

### 2.4. Cstb Promotes Migration of Stromal Cells in Mouse

A previous report stated that *Cstb* was involved in the metastasis of tumor cells [16]. Non-decidualized stromal cells have a lower migratory capacity than decidualized cells. This prompted us to study the role of *Cstb* in the migration of endometrial stromal cells. We knocked down the gene of *Cstb* through siRNA and then scraped a trace in the cultured mouse uterine stromal cells, using the tip of a micropipette. The control cells migrated and almost healed after 72 h. However, the cell migration rate of the si*Cstb* group was much slower than that of the control group (Figure 4A,B). The interference efficiency was tested using RT-qPCR, and this showed that the interference efficiency of si*Cstb*-1 transfection treatment was most obvious, reducing the expression of *Cstb* by above 50% (Figure 4C), and so it was used for the subsequent experiment.

In addition, a transwell experiment was performed. The cells were first spread on the upper layer of the chamber, and then siRNA was used to inhibit the expression of *Cstb* in mouse stromal cells. After 12 h, the cells in the lower layer of the chamber were stained with DAPI and observed under a fluorescence microscope. It could be clearly observed that the number of migrating cells in the control group was greater than in the si*Cstb* group (Figure 5A,B). These results all indicated that *Cstb* regulated the migration of stromal cells.

### 2.5. Function of Cstb during Mouse In Vitro Decidualization

Since *Cstb* is strongly expressed in decidual tissue, we examined whether *Cstb* played a role in decidualization. After induction of decidualization in mouse endometrial primary stromal cells using 17 β-estradiol (10 nM) and progesterone (1 μM), the expression of *Dtprp* in the si*Cstb* group significantly reduced (Figure 6A), and the *Cstb* level in the si*Cstb* decidualization group was pronouncedly lower than in the control decidualization group (Figure 6B). This confirmed the important role of *Cstb* in mouse decidualization.

### 2.6. Cstb Expression Is Associated with Angiogenesis in the Decidual Cells

To further explore the molecular mechanism by which *Cstb* regulates decidualization, we examined gene expression profiles in endometrial decidual cells by comparing the RNA-Seq of the primary stromal cells from the control and si*Cstb* groups under in vitro decidualization. Heat map analysis showed that gene expression levels were significantly different between si*Cstb* group and the control group (Figure 6C). As shown in the volcano plot, RNA-seq analysis revealed a large number of changed genes (Figure 6D). After comparing differentially expressed genes in the two groups, we found that significant changes could be observed in molecules associated with angiogenesis regulatory pathways (Figure 7A).

The uterus proliferated and differentiated to form an SDZ full of blood vessels on day 6 during early pregnancy in mice, and our results showed that *Cstb* was abundantly distributed in the SDZ on day 6 of pregnancy. Therefore, we localized the avascular primary decidual zone marker molecule ZO1 and *Cstb* by immunofluorescence, and the results showed that the locations of two of them were complementary (Figure 7B). This further suggested that *Cstb* might be involved in the process of angiogenesis in the decidual region of the uterus.

The RNA-seq data showed that *Angptl7*, which promotes angiogenesis, was one of the most down-regulated genes in the dc si*Cstb* group (Appendix A). Compared with the control group, the expression of *Angptl7* was significantly down-regulated in the si*Cstb* group (Figure 7C). *Angptl7* We detected the expression of *Angptl7* after embryo implantation in the mouse uterus by in situ hybridization. The results showed no *Angptl7* mRNA expression in the avascular primary decidual zone, but strong expression in other decidual areas on day 6 (Figure 7D), which suggested that *Angptl7* might mediate the regulation of *Cstb* in the process of decidualization.

## 3. Discussion

The establishment of pregnancy is a complex process that requires the coordination of several key steps. Estrogen, progesterone, and their receptors are responsible for directing and coordinating these events, which are important for the successful establishment of early pregnancy. On day 1 of pregnancy, estrogen stimulates the proliferation of endometrial epithelial cells. On days 2–3, the endometrium prepares for implantation under the action of the corpus luteum, which produces an increasing amount of progesterone. On day 4 of gestation, the progesterone is at a high level, while the surge of estrogen induces the expression of many genes critical for uterine receptivity, marking the beginning of the receptivity window. The mouse uterus receives blastocyst implantation on day 4 of gestation, after which differentiation of endometrial stromal cells is triggered, and the decidualized area of the matrix surrounding the implantation chamber is gradually expanded [1,2].

This study researched the induction of *Cstb* by estrogen via ERα in the decidual region of pregnant mice. In rodents, most of the molecular changes associated with post-implantation decidua are closely related to the continuous supply of progesterone [20]. However, studies have shown that non-exogenous estrogen is also required for decidualization. After embryo implantation, the uterus can synthesize estrogen by itself, and blocking the process can greatly damage the formation of the decidual region [6]. These factors all demonstrate that certain molecules induced by estrogen are also very important for the initiation of the decidualization process. In this study, blocking estrogen synthesis in mice also inhibited the expression of *Cstb* in the decidual region. Therefore, it could be considered that endogenous estrogen induces the expression of *Cstb*.

Both *Cstb* mRNA and protein were induced in the subepithelial stroma at the implantation site of the mouse uterus on day 5 of gestation and gradually increased from day 6 to day 8 of pregnancy. The signal expression area continued to expand, with the same trend as the decidualized area. In the decidual zone of the artificially induced decidualized uterus, the expression was remarkable. Following the onset of embryonic implantation and invasion, various distinct morphological and qualitative changes occur in the tissues adjacent to the implanted embryo, including ECM remodeling and consequent changes in cell migration [21,22,23,24]. Trophoblast invasion involves endometrial ECM hydrolysis and migration of decidual cells. These processes are coordinated, and various factors can be regulated to balance each other. Cathepsins play an important role in regulating the interactions among the embryo, the endometrium, and the invasion of trophoblast cells. However, their enzymatic activities are under the control of endogenous inhibitors [25,26]. As an inhibitor of many proteases, cystatin plays a key role in inflammation.

In the present study, it was found that interfering with *Cstb* expression in mouse endometrial primary stromal cells could reduce the migratory ability of stromal cells. These observations were also consistent with a function of *Cstb* in the migration of some cancer cells. It has been demonstrated that overexpression of *Cstb* in pancreatic and liver cancer cells can promote their migratory and metastatic abilities [16,27]. It has also been suggested that Cathepsins may promote the extravasation of cancer cells by enhancing the activity of metal matrix proteinases [28]. Therefore, the balance and coordination of these enzymes and enzyme inhibitors can provide a suitably dynamic micro-environment for the implantation and development of early-gestation embryos. In addition, studies have shown that the proliferation and invasion of embryonic trophoblast cells also require the movement of maternal endometrial stromal cells [29], and decidual cells have a stronger migrating ability compared with non-decidual cells [30]. Early decidual angiogenesis after implantation is essential for normal pregnancy development. The newly formed decidual vasculature is the first element in the developing embryo to exchange nutrients with the mother, prior to the functional maturation of the placenta [31]. In this study, the RNA-seq results and fluorescence detection of *Cstb* and ZO1 localization indicated that *Cstb* might be involved in the angiogenesis process in the decidual area of the mouse uterus.

Endometrial cancer is one of the most common cancers in women worldwide, and some studies showed that *Cstb* expression was significantly upregulated in endometrial adenocarcinoma tissues compared with normal tissues [32]. In addition, in the tissue of patients with recurrent miscarriages, the concentration of *Cstb* was significantly higher than in control individuals [18]. Our data suggest that *Cstb* may mediate estrogen regulation of angiogenesis during decidualization. These findings may contribute critical insights into therapeutic strategies for women’s diseases.

Angptl proteins are known to function as a factor in angiogenesis. In earlier studies, people found that *Angptl7* was normally expressed in tissues without blood vessels, suggesting that it might be a negative regulator of angiogenesis [33]. However, some studies in recent years have produced conflicting results. *Angptl7* has also been reported to have strong pro-angiogenic activity in vivo [34]. Our study showed *Angptl7* was mainly distributed in the SDZ, which might suggest that *Angptl7* plays a role during decidualization.

In summary, our study describes the expression, regulation, and function of *Cstb* during decidualization in mice. Due to the strong expression of *Cstb* in the decidual region, it can be used as a potential target for contraception, and endometrial-related diseases, such as endometrial cancer and recurrent miscarriages. Since *Cstb* is induced by estrogen and may be related to endometrial cancer cell metastasis, it may be a new entry point for the alleviation and treatment of these diseases.

## 4. Materials and Methods

### 4.1. Animals and Treatment

CD1 mature mice were housed in an SPF animal room in a controlled environment (22–24 °C and 60–70% relative humidity) and were maintained on a light/dark cycle (12 h light/12 h dark). Male ERα heterozygous KO mice on C57Bl/6J were obtained from Jackson laboratory and crossed with female CD1 mice to generate ERα knockout mice in CD1 background [35,36]. All animal procedures have been approved by the Animal Care and Use Committee of South China Agricultural University (No. 2021f108). Female mice were mated with fertile or vasectomized males in the afternoon to induce pregnancy or pseudopregnancy. The date of detection of the vaginal plug was defined as day 1 of pregnancy. Pregnancy on days 1–4 is confirmed by recovering the embryo from the oviduct and uterus. Intravenous injection of Chicago blue dye solution (Sigma-Aldrich, St. Louis, MO, USA) was used to determine the implantation site on days 5 and 6 of pregnancy.

To induce artificial decidualization, sesame oil (10 μL, Sigma-Aldrich, St. Louis, MO, USA) was injected into one uterine horn on day 4 of pseudopregnancy while the noninjected contralateral horn served as a control. Deciduoma was observed on day 8 of pseudopregnancy. To determine the effect of steroid hormones on *Cstb* expression, all of the experiments were conducted 2 weeks after ovariectomy. Mice were given a single injection of 17β-estradiol (100 ng per mouse, Sigma-Aldrich, St. Louis, MO, USA), progesterone (1 mg per mouse, Sigma-Aldrich, St. Louis, MO, USA), propyl pyrazole triol (PPT, 250 ng per mouse; Tocris, Bristol, UK), a selective ERα agonist, or ICI182780 (ICI, 100 ng per mouse; Sigma-Aldrich, St. Louis, MO, USA), an ER antagonist. To further confirm whether *Cstb* expression was ERα-dependent, wild-type (WT) or ERα-knockout (ERαKO) mice were rested for 2 weeks after ovariectomy and then treated with the 17 β-estradiol injection. Uterine was harvested 24 h after injection.

### 4.2. RNA Extraction and Real-Time Quantitative Polymerase Chain Reaction (RT-qPCR)

RT-qPCR was performed as previously described [37]. Tissues or cells were collected and immediately stored at −70 °C. Total RNA was isolated using a protocol based on TRI reagent (Sigma-Aldrich, St. Louis, MO, USA), followed by quantification and quality assessment according to the manufacturer’s instructions. Furthermore, cDNA was synthesized using the PrimeScript Reverse Transcriptase Kit (TaKaRa, Mountain View, CA, USA), according to the manufacturer’s instructions. The conditions used for RT-qPCR were as follows: 95 °C for 10 s followed by 39 cycles of 95 °C for 5 s and 60 °C for 34 s. All reactions were run in triplicate. The ΔΔCt method was employed to determine relative changes in gene expression compared to *Rpl7*.

### 4.3. In Situ Hybridization

In situ hybridization was performed as previously described [38]. In brief, total RNAs were isolated from the mouse uterus on day 8 of pregnancy and amplified with *Cstb* primers after reverse transcription. The amplified gene fragment was cloned into the pGEM-T plasmid (Promega) and sequenced. For template preparation, the plasmids were amplified with primers for SP6 and T7. Digoxigenin-labeled antisense or sense complementary RNA probes were transcribed in vitro using a digoxigenin RNA labeling kit (Roche Applied Science, San Francisco, CA, USA), according to the manufacturer’s instructions. Primers used for in situ hybridization were listed in Table 1. Frozen uterine sections (10 μm) were fixed in 4% paraformaldehyde solution for 1 h. They were hybridized at 55 °C for 16 h. The sections were then incubated in the alkaline phosphatase-conjugated anti-digoxigenin antibody (1:5000; Roche Applied Science, Penzberg, Germany). The positive signal was identified using a buffer containing 0.4 mM 5-bromo-4-chloro-3-indolyl phosphate (Amresco, Solon, OH, USA) and 0.4 mM nitro blue tetrazolium (Amresco, Solon, OH, USA) as a dark brown color. All of the sections were counterstained with 1% methyl green.

### 4.4. Isolation and Treatment of Endometrial Stromal Cells

Endometrial Stromal Cells were isolated and treated as previously described [39]. Uteri collected from day 4 of pregnancy mice were dissected and digested with 1% (*w*/*v*) trypsin (Amresco, Cleveland, OH, USA)) and 6 mg/mL dispase (Roche Applied Science, Penzberg, Germany) in Hanks’ balanced salt solution (HBSS, Sigma, St. Louis, MO, USA) for 1 h at 4 °C, 1 h at room temperature and 10 min at 37 °C to remove epithelial cells. The remaining pieces of tissues were digested with 0.15 mg/mL collagenase I (Invitrogen, Houston, TX, USA) in HBSS at 37 °C for 30 min, followed by vigorously shaking fifty times. All liquids were then sieved through a 70 μm strainer, and then centrifuged to collect stromal cells. Cells were resuspended and grown in DMEM/F-12 (Sigma-Aldrich, St. Louis, MO, USA) medium with 10% charcoal-treated fetal bovine serum (cFBS, Biological Industries, Beit-Haemek, Israel).

To induce decidualization in vitro, mouse endometrial stromal cells were treated with 10 nM of 17β-estradiol plus 1 μM of progesterone for 48 h. For further studies, the siRNA kit which included one scramble control and three interference fragments targeting *Cstb* was synthesized by Ribobio Co., Ltd. (Guangzhou, China). A random RNA sequence not specifically for any specific genes (scramble) was used as negative control (NC). Three targeting siRNAs (50 nM) were used to transfected into the stromal cells, the relative mRNA levels of *Cstb* were detected. Among them, siRNA, which has the most remarkable interference efficiency, was used for subsequent research. The stromal cells were transfected with 50 nM siRNA of *Cstb* using Lipofectamine 3000 (Invitrogen, Waltham, MA, USA) according to the manufacturer’s protocol. After 6 h, the cells were induced in vitro decidualization. The sequences of targeting siRNAs are shown in Table 1.

### 4.5. Immunohistochemistry

Immunohistochemistry was performed as previously described [40]. Paraffin sections were deparaffinized with Xylene and alcohol gradient solutions, then washed with deionized water. Sections were repaired in a microwave oven for 10 min in sodium citrate buffer (pH 6.0) and then soaked in 3% H_2_O_2_ for 15 min to remove endogenous horseradish peroxidase. After blocking with 10% horse serum (Zhongshan Jinqiao, Beijing, China) in a 37 °C oven for 30 min, the sections were incubated with mouse anti-*Cstb* (1:800, Santa Cruz, sc-166561)diluted in phosphate-buffered saline (PBS) overnight in a 4°C refrigerator. They were then washed and incubated with biotin-labeled goat anti (mouse IgG) antibody (Zhongshan Jinqiao, Beijing, China) for 30 min, followed by streptavidin-HRP complex (Zhongshan Jinqiao, Beijing, China) for 30 min. The positive signals were shown as reddish-brown using the DAB Horseradish Peroxidase Chromogenic Kit (Zhongshan Jinqiao, Beijing, China) according to the manufacturer’s protocol. The sections were counterstained with hematoxylin.

### 4.6. Immunofluorescence

Immunohistochemistry was performed as previously described [41]. Briefly, frozen sections were fixed in 4% paraformaldehyde solution for 10 min and then soaked in 0.1% Triton X-100 in PBS for 15 min. After blocking with 5% donkey serum (Zhongshan Jinqiao, Beijing, China) in a 37 °C oven for 60 min, the sections were incubated with mouse anti-*Cstb* (1:500, Santa Cruz, sc-166561), ZO1 (1:200, Cell Signaling Technology, Boston, MA, USA, #13663) diluted in PBS overnight in a 4 °C refrigerator. The primary antibody was washed off with PBST the next day. The sections were incubated with FITC-conjugated secondary antibody (Jackson ImmunoResearch Laboratories, West Grove, PA, USA) for 30 min at 37 °C, and nuclei were counterstained with 4′,6-diamidino-2-phenylindole dihydrochloride (DAPI, Sigma-Aldrich, St. Louis, MO, USA). Fluorescence microscopy was used to observe and take pictures.

### 4.7. Migration Analysis by Millicell Transwell Chamber

Cells were plated on the upper Millicell transwell chamber with a density of 2 × 10^5^ cells in 24-well plates, and 400 μL of DMEM/F-12 medium without cFBS was added to the lower chamber and incubated at 37 °C in a humidified atmosphere of 5% CO_2_ and 95% air. The stromal cells were transfected with siRNA of *Cstb* using Lipofectamine 3000 (Invitrogen, Waltham, MA, USA), according to the manufacturer’s protocol. After the transfection was over, the lower liquid was converted to 400 μL of medium containing 10% fetal bovine serum (Biological Industries, Beit-Haemek, Israel) and then incubated at 37 °C for 12 h. Then, the filters were fixed with 4% paraformaldehyde solution and stained with DAPI. The cells on the upper surface of the filters were removed by wiping them with cotton swabs. The cells that had migrated to various areas of the lower surface were manually counted under a fluorescence microscope.

### 4.8. Wound Healing Assay

Cells were seeded in a six-well plate and incubated at 37 °C until they reached 80% confluence. A wounding line was scratched with a 1000 μL pipette tip, and the dead cells were washed with PBS. Then the stromal cells were transfected with *Cstb* siRNA using Lipofectamine 3000 (Invitrogen, Waltham, MA, USA) according to the manufacturer’s protocol. Then, DMEM/F-12 medium without cFBS was added to each well. The migrating cells were monitored using an inverted microscope. Images were taken in four randomly selected fields at 0, 24, 48, and 72 h. The distance of migrating cells was counted based on the captured images using ImageJ software.

### 4.9. RNA-Seq and Data Analysis

The stromal cells were cultured in 6-well cell culture plates and transfected with indicated siRNA after decidualization and decidualization treatment, respectively. Total RNA of these cells was extracted with TRI reagent (Sigma-Aldrich) and used for RNA-seq analysis, conducted by BGISEQ (Chinese BGI). The data obtained by RNA-seq in the conversion of different groups of genes were used to reflect the expression of mRNA. All genetic data were analyzed using software R 4.1.1. EdgeR software was used to perform differential expression analysis of all gene data from different groups to obtain Fold Changes, Log2 (Fold Changes), and *p*-values for each gene, and to screen for difference genes (DEGs) based on Log2 (Fold Changes) >0.25 or <−0.25 and *p*-values < 0.05. GO was performed using R 4.1.1.

### 4.10. Statistical Analysis

All experiments were repeated at least three times, independently. All data represent the mean ± standard deviation (SD). Statistical analyses were performed using the student’s *t*-tests *and* analysis of variance (ANOVA) with the Newmann–Keuls test, and *p* < 0.05 was considered statistically significant.

## Figures and Tables

**Figure 1 ijms-24-00302-f001:**
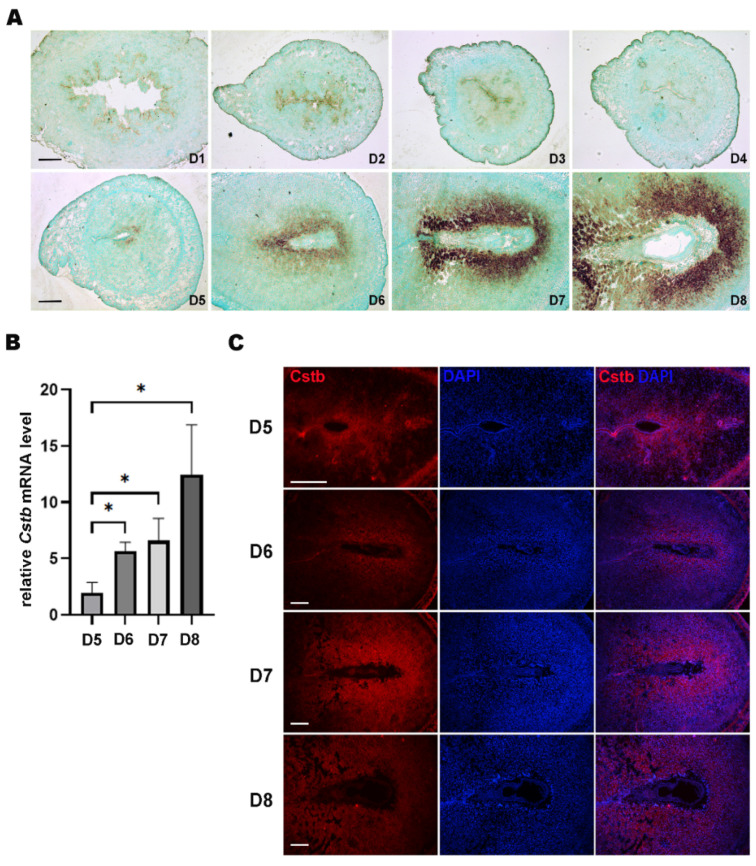
Expression of *Cstb* in the mice uteri during early pregnancy. (**A**) In situ hybridization showing the expression of *Cstb* in mice uteri from days 1 to 8 of pregnancy. Bar = 300 μm. (**B**) RT-qPCR of *Cstb* mRNA level in the mice uteri from days 5 to 8 of pregnancy. (**C**) Immunofluorescence of *Cstb* protein from days 5 to 8. The sections were stained with *Cstb* antibody (red), and the nuclei were counterstained using DAPI (blue). Bar = 300 μm. * *p* < 0.05.

**Figure 2 ijms-24-00302-f002:**
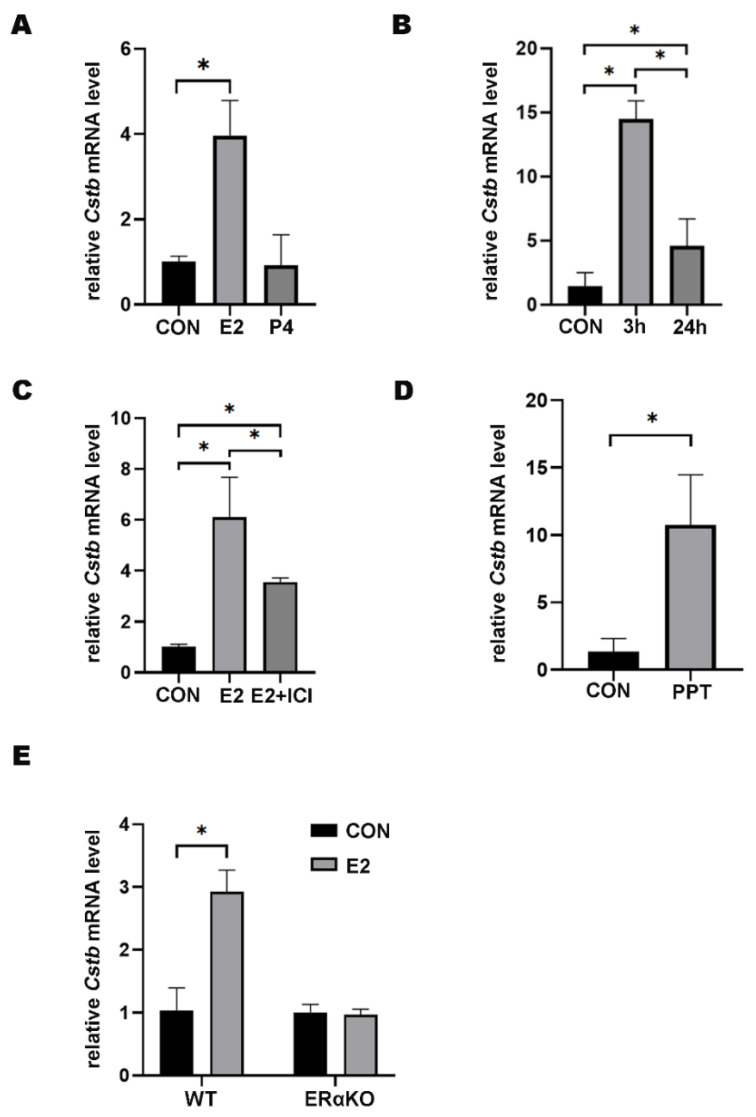
Regulation of Estrogen on *Cstb* Expression. (**A**) RT-qPCR analysis of *Cstb* expression in the uterus of ovariectomized mice treated with progesterone (1 mg per mouse) and estrogen (100 ng per mouse) for 24 h subcutaneously. (**B**) *Cstb* expression in the uterus of ovariectomized mice treated with estrogen was detected by RT-qPCR after 0 h, 3 h, 24 h, respectively. (**C**) Effect of ICI182780 on *Cstb* expression in ovariectomized mice after 3 h of estrogen stimulation. (**D**) RT-qPCR analysis of *Cstb* levels after ovariectomized mice were treated with PPT for 3 h. (**E**) Ovariectomized ERα knockout mice and ovariectomized wild-type mice were injected with sesame oil or estrogen, respectively, for 3 h, and the expression of *Cstb* was detected by RT-qPCR. CON, control; ERαKO, ERα knockout mouse; WT, ERα wild-type mouse; E2, estrogen; P4, progesterone; ICI, ICI182780; PPT, propyl pyrazole triol. * *p* < 0.05.

**Figure 3 ijms-24-00302-f003:**
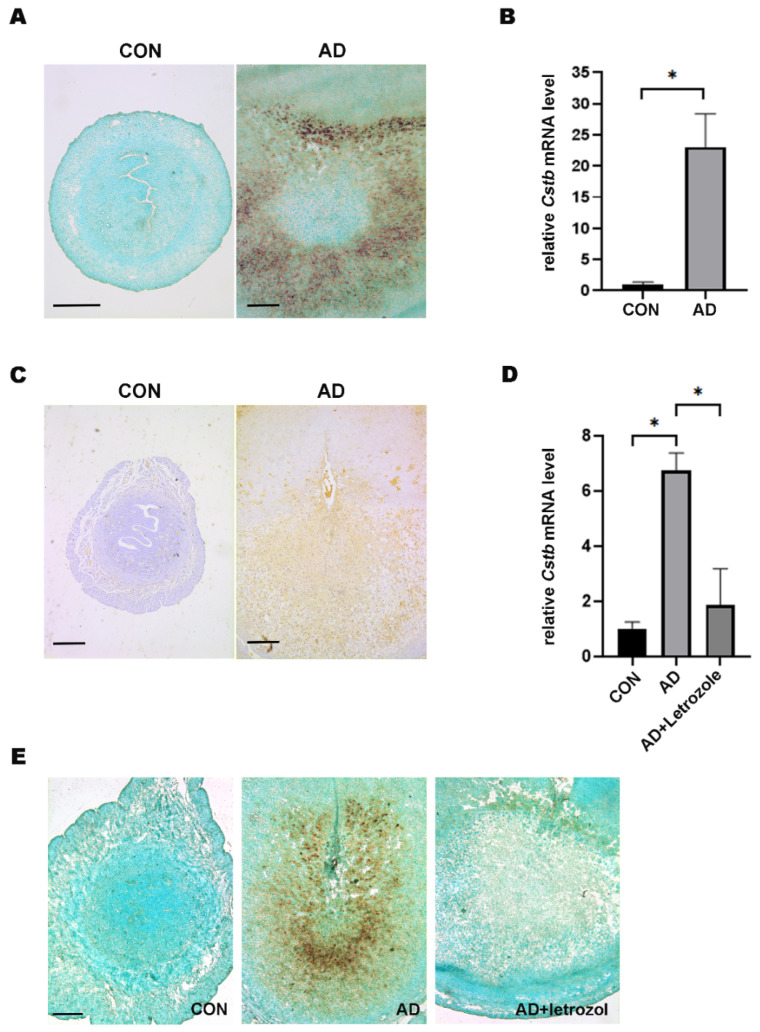
Endogenous estrogen in the womb is involved in the regulation of *Cstb* expression during decidualization. (**A**) The uteruswas collected on day 8 by injecting oil into the pseudopregnant uterus on the morning of day 4. *Cstb* mRNA signal was detected by in situ hybridization. Bar = 300 μm. (**B**) The expression of *Cstb* was detected by RT-qPCR. (**C**) The expression of *Cstb* was detected by immunohistochemistry. Bar = 300 μm. (**D**) The expression of *Cstb* was detected by RT-qPCR. (**E**) The expression of *Cstb* was detected in situ hybridization. CON, control; AD, Artificial decidualization; Letrozole: aromatase inhibitor. Bar = 300 μm. * *p* < 0.05.

**Figure 4 ijms-24-00302-f004:**
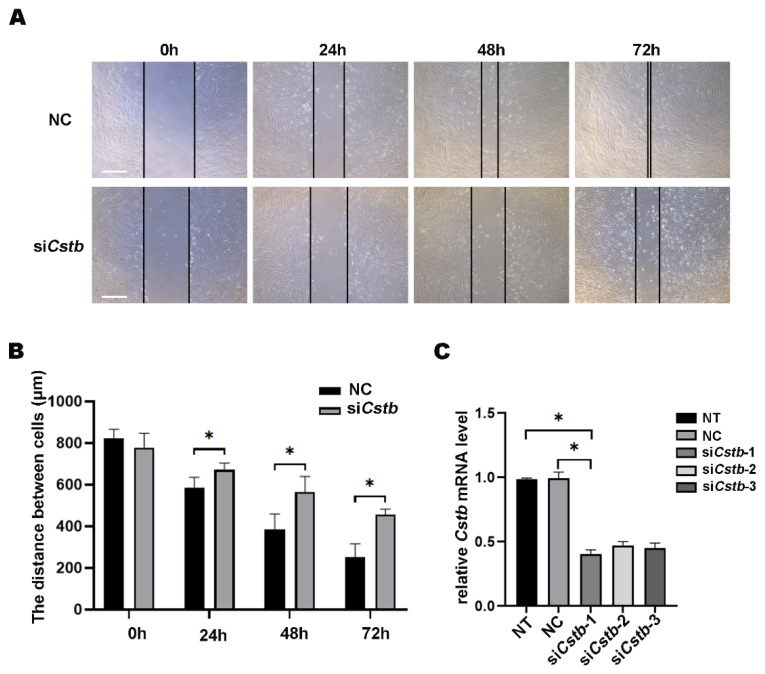
*Cstb* regulates the migration of mouse uterine stromal cells. (**A**) Stromal cells treated with siRNA were examined for wound closure 72 h after a wound scratch was made. Live cell images were captured at 0 h, 24 h, 48 h, 72 h. Bar = 400 μm. (**B**) Distance between stromal cells treated siRNA was examined. (**C**) The expression of *Cstb* in stromal cells treated with siRNA was detected by RT-qPCR. NT, no-treatment control; NC, negative control; si*Cstb*, siRNA of *Cstb*. * *p* < 0.05.

**Figure 5 ijms-24-00302-f005:**
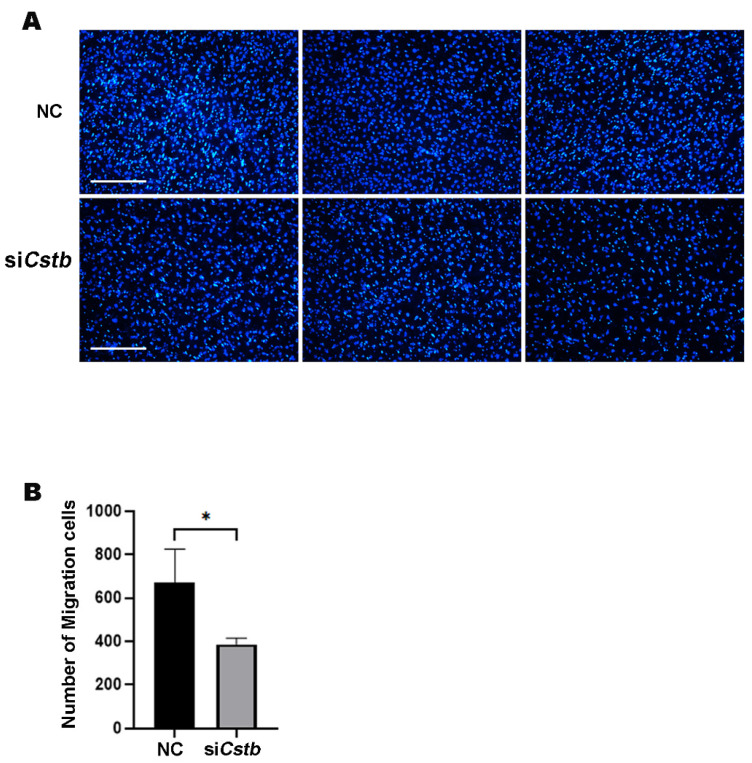
*Cstb* promotes the migration of mouse uterine stromal cells. (**A**) Stromal cells were treated with ordinary medium and transfected with siRNA for 12 h. Cells migrating to the lower layer of the transwell were stained with DAPI (blue) by immunofluorescence. Bar = 300 μm. (**B**) The number of stromal cells was examined. NC, negative control; si*Cstb*, siRNA of *Cstb*. * *p* < 0.05.

**Figure 6 ijms-24-00302-f006:**
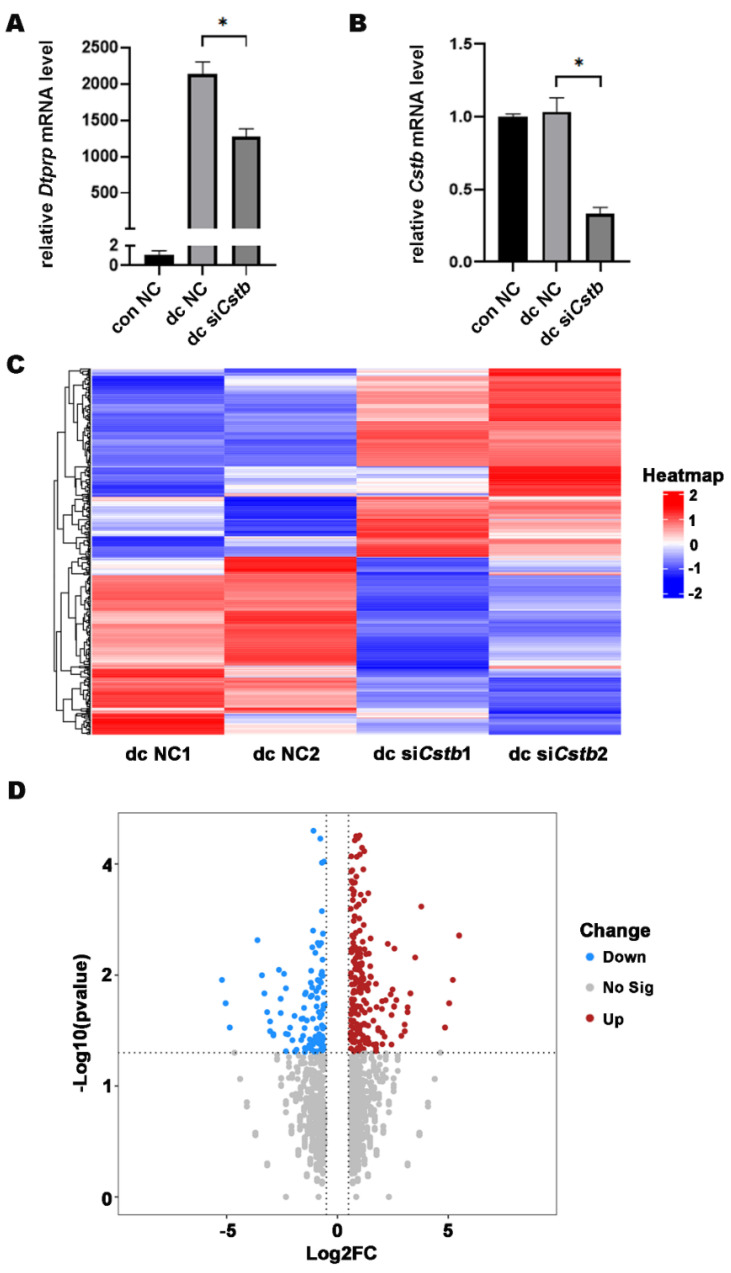
*Cstb* mRNA expression during in vitro decidualization. (**A**) Estrogen (10 nM) and progesterone (1 μM) were used to induce decidualization of primary mouse endometrial stromal cells for 48 h after RNA interference in mouse stromal cells for 6 h. The expression of *Dtprp* was detected by RT-qPCR. * *p* < 0.05. (**B**) RT-qPCR analysis on the effect of *Cstb* siRNA on *Cstb* mRNA level in stromal cells. (**C**) Heat map showing hierarchical clustering of gene expression in dc NC group and dc si*Cstb* group. Differentially expressed genes were screened according to Log2 (Fold Changes) > 0.25 or <−0.25 and *p*-value < 0.05. Red and blue depict high and low levels of gene expression, respectively. (**D**) Volcanic maps show differential expression of genes in the dc NC group versus the dc si*Cstb* group. The upregulated gene (red); the downregulated gene (blue). NC, negative control; si*Cstb*, siRNA of *Cstb*; dc, in vitro decidualization.

**Figure 7 ijms-24-00302-f007:**
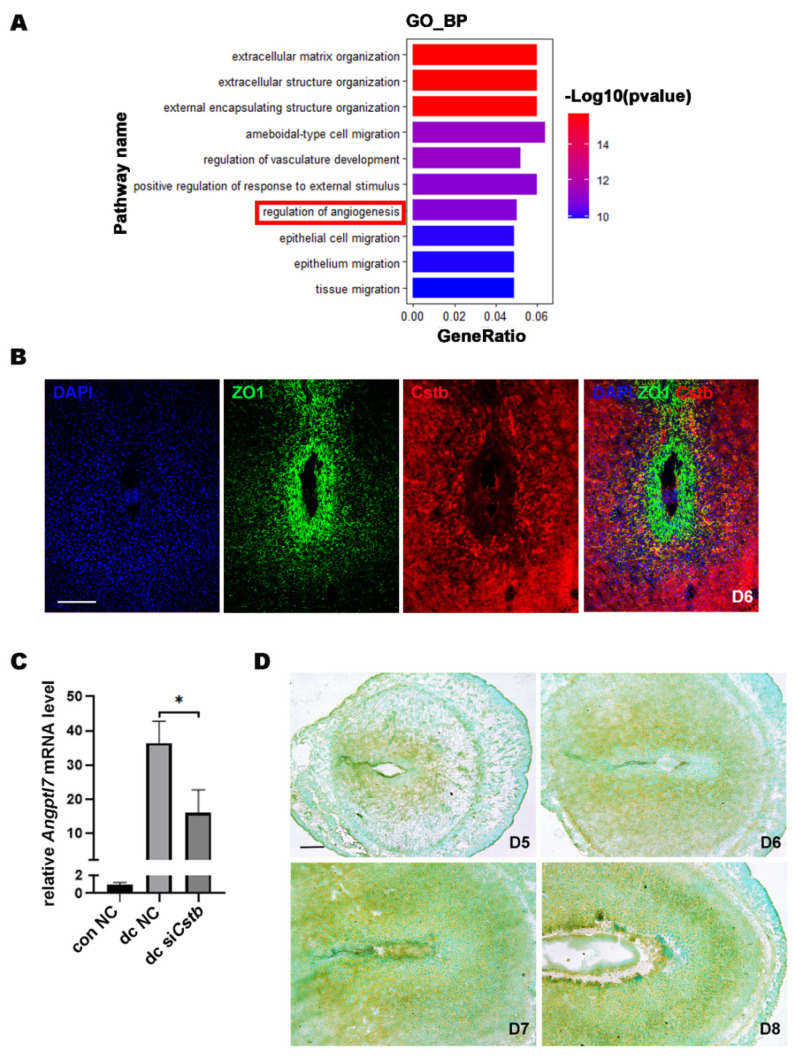
*Cstb* expression is associated with angiogenesis in mouse uterine stromal cells*Angptl7*. (**A**) Gene ontology (GO) functional classification of the DEGs. (**B**) The expressions of *Cstb* and ZO1 were detected by immunofluorescence in the uterus on day 6. Bar = 300 μm. (**C**) The expression of *Angptl7* in stromal cells of the con NC group, dc NC group, and dc si*Cstb* group was detected by RT-qPCR. (**D**) The expression of *Angptl7* in mouse uteri from days 5 to 8 of pregnancy was detected by in situ hybridization. Bar = 300 μm. * *p* < 0.05. NC, negative control; si*Cstb*, siRNA of *Cstb*; dc, in vitro decidualization.

**Table 1 ijms-24-00302-t001:** Primers and siRNA sequences used in this study.

Gene Name	Primer Sequences	Size (bp)	Application	Accession Number
*Cstb*	CCTAGTTGGATCTGTCTTCA AGCCACTATCTGTCTCTTG	201	ISH	NM_007793.3
*Cstb*	AGGTGAAGTCCCAGCTTGAAT GTCTGATAGGAAGACAGGGTCA	196	RT	NM_007793.3
*Dtprp*	*AGCCAGAAATCACTGCCACT* *TGATCCATGCACCCATAAAA*	*119*	*RT*	*NM_001289919.1*
*Angptl7*	TAAACGCAAGACACAGCTCAA TGCATGATGTCAATCTGGTTGT	237	ISH	NM_001039554.3
*Angptl7*	TGACTGTTCTTCCCTGTACCA CAAGGCCACTCTTACGTCTCT	158	RT	NM_001039554.3
*Rpl7*	GCAGATGTACCGCACTGAGATTCACCTTTGGGCTTACTCCATTGATA	129	RT	NM_011291.5
*Cstb*-1	CAGCTTGAATCGAAAGAAA		siRNA	
*Cstb*-2	GCACTTGAGGGTGTTTCAA		siRNA	
*Cstb*-3	CCTATCAGACCAACAAAGA		siRNA	

## Data Availability

All the data generated in this study are included in this manuscript.

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
