# Peer review of "Expression and Regulation of a Novel Decidual Cells-Derived Estrogen Target during Decidualization"

_ijms, 2022, doi:10.3390/ijms24010302_

Round 1

Reviewer 1 Report

Comments on manuscript IJMS-2032021

entitled

“Expression and Regulation of a Novel Decidual Cells-derived Estrogen Target during decidualization” authored by Lu et al..

In their manuscript the authors present a principally interesting set of experimental data, which indicate a role for estrogen induced CstB during decidualization of the mouse uterus. Part of the experiments seem to be well designed and much content of the results and methods sections is adequately described.

However, the manuscript suffers from several weaknesses and major issues, which must be eliminated or clarified, respectively, before the manuscript can be accepted.

Major points

1 Throughout the manuscript (with only one exception in fig. 6), the authors give no information about replicates. It is absolutely mandatory to include the numbers of biological replicates for each experimental group, at least in the figure legends. Technical replicates or different images from the same specimen cannot replace biological replicates.

2 The authors often indicate p-values and imply that they indicate significant differences. However, they nowhere describe the statistical tests used (again with the exception of the experiment summarized in fig. 6, which obviously was done by another party, which is not revealed in line 402). Therefore, clearly describe statistical tests used for comparisons and indicate them in the legends.

3 One major point of concern are the animals used. In line 297 it is stated that CD1-mice were used. But what was the genetic background of the ERaKO-mice? How where they generated or where they were from? Give reference.

4 The siRNA-experiments throughout lack the experimental group treated with the adequate non-template control siRNA, which is absolutely mandatory according to guidelines for siRNA-experiments. The fact that you see an effect with one siRNA alone proves nothing.

The respective description in methods does not give sequences, concentrations and so on.

5 Information about critical reagents is missing or sometimes erroneous. For antibodies at least the order-numbers are necessary. Did you really use 1 mg progesterone per mouse (i.e. roughly 100 µM)? Other examples: propyl pyrazole triol is not listed at Sigma-Aldrich; DAPI is neither 6-diamino-2-phenylindole (page 17) nor 4,6-diamidino-2-phenylindole but 4’,6-diamidino-2-phenylindole;……. Please, carefully control the manuscript for other imprecisions.

6 Up to reference 18, the manuscript is well referenced. However, the next references (at least 19-23 in the list) do not prove what is claimed at their first mentioning in the text. Something has gone wrong with the order of references. It is a nuisance for reviewing. Therefore, I refused to check references above number 23.

In addition, the authors should uniformly present/format the references in the bibliography (e.g. full versus abbreviated journal titles; why two-fold numbering?). At least these references include typos or lacking information: 15, 30.

Referencing of the methods section is poor – you surely did not invent all these methods.

7 The introductory part of the abstract is excessive (roughly 60%), whereas the methods used are almost not mentioned.

8 The English has to be improved. Although much of the text is fine, there are sentences where one has to guess what is really meant.

9 A table of differentially regulated genes in the RNAseq-experiment is missing. Does it really make sense to calculate p-values for an experiment, where n=2 for both experimental groups?

Other points

-       Use of capital letters in (sub)titles is not consistent.

-       In many instances abbreviations used in figures are not explained in the legends (e.g. CON, NC & many more).

-       -There are a considerable number of typos and strange expressions.

-       Lines 42-47: Is ref 7 really the paper, which defined the terms “primary decidua(l zone)” and “secondary decidual zone” – both terms are used exactly once in that paper.

-       Line 84: during days 5-8.

-       Line 87: Regulatory role of Estrogen…

-       Lines 91-93: Estrogen “induces” no downstream molecules….

-       Lines 96-97: Statement is incorrect, as half of the estrogen effect is still detectable.

-       Fig. 2B and text: There is no explanation for the dramatic loss of CstB-expression after 24 h subcutaneous E2-injection (in relation to 3 h). This does not really fit to the results from Fig. 1.

-       Fig. 2C: Time?

-       Fig. 4: Poor image quality. Some lines seem to be set rather arbitrarily.

-       Fig. 6B: Why has decidualization no effect on CstB-expression? This is in disagreement with results shown previously.

-       Lines 203-205: Statement is not backed by data given in the manuscript.

-       Lines 209-210: This statement does not make sense!

-       Fig. 7B: Does not show co-localization – expression of both proteins is almost perfectly separated!

-       Lines 242-245: As the aromatase inhibitor was given systemically, it cannot be claimed that the blocking aromatase in the uterus is responsible for the observed effects. What can be claimed is that estrogen synthesis is necessary.

-       Lines 290-294: This is pure speculation.

-       Tab. 1: Rpl8a2 ?

-       Lines 350-354: Formatting. – Is decidualization possible in a cell culture monolayer?

-       Lines 408-409: There was no material provided.

-       List of abbreviations seems to be copy-pasted from another manuscript, as many abbreviations in the list are not used in the manuscript and on the other hand, many abbreviations used in the manuscript are not listed here.

In summary, a very careful checking of the whole manuscript is necessary.

Reviewer 2 Report

Lu et al study wants to focused on the Expression and Regulation of a Novel Decidual Cells-derived 2 Estrogen Target during decidualization, and in particular they focused their attention on Cystatin B like the main actor during mouse decidualization.
In my opinion the topic is original and relevant  to deepen the subject in order to evaluate the role of Cystatin B also in pathological conditions. The methodologies used and the results obtained fully describe the questions that the authors ask themselves. The references are appropriate and event the conclusion that address the main question posed.

Only few questions:

1.How large is the sample? Please better describe this point.

2. Figure 4 A the wound healing is not so visible from that figure. Please try to that better figure with a different magnification.

3. In the discussion take a space to describe the importance of your discovery in more detail. For example, why is important to go deeper in your study for pathological condition like endometriosis or other disease?

Round 2

Reviewer 1 Report

Comments on manuscript IJMS-2032021 – version 2

entitled

“Expression and Regulation of a Novel Decidual Cells-derived Estrogen Target during Decidualization” authored by Lu et al..

The authors corrected a lot of minor and major issues in their revised manuscript.

However, there are still some points to be corrected and one major issue regarding original point 4 to be resolved:

Point 3/Response 3: … We generated the ERαKO-mice as follows:

Male ERα heterozygous KO mice on C57Bl/6J were obtained from Jackson laboratory and crossed with female CD1 mice to generate ERα knockout mice in CD1 background (PMID: 35409055).

 Please add this information and reference to methods (4.1).

Point 4: The siRNA-experiments throughout lack the experimental group treated with the adequate non-template control siRNA, which is absolutely mandatory according to guidelines for siRNA-experiments. The fact that you see an effect with one siRNA alone proves nothing.

The respective description in methods does not give sequences, concentrations and so on.

Response 4: Thank you very much for your suggestions. Our sentence in the results is ambiguous. NC refers to non-template control siRNA. The concentration of the siRNA we used was 50nM, and we have listed the sequences of the siRNA in table 1.

 There is still no sufficient information for a reliable siRNA-experiment. This would need a ‘no treatment control’, a ‘non-template control’ and at least two ‘targeting siRNAs’. Due to the very short coding sequence of CstB it might be problematic to fulfill the last requirement (≥ two targeting siRNAs). However, the other requirements are easy to meet and are absolutely necessary to avoid false interpretation of data. Therefore, at least one essential control is still lacking.

There should be at least one reproducible (biological replicates) set of data, which compares the effect of ‘target siRNA’ with the TWO INDEPENDENT controls (e.g. a setup as Fig. 4C, but containing both control groups). Only then can be validated, whether the observed effect is really specific for the target knock-down or whether it is merely an effect resulting from some un-specific effect on general mRNA-expression. (The con NC group in Fig 6 does not fulfill that criteria).

The sequence for the ‘non-template (or scrambled) control’ should be also given in the table.

Point 6: …..Referencing of the methods section is poor – you surely did not invent all these methods.

Response 6: …. And we have added the references to the methods section.

 As the mentioned techniques are used for decades, the self-citations (references 35-39) are inappropriate. The descriptions give information about the details of the applied method. The references should refer to the inventors of the basic methods!

Perhaps I missed it, but an explanation for ‘NC’ is still not given in the text.

General: The use of ‘was’ and ‘were’ is still not correct at many locations.

And some sentences still sound a little bit weird, e.g. 158-159: Use the tip of a micropipette to scrape a trace in the cultured mouse uterine stromal cells and then knock down the gene of Cstb in mouse stromal cells through siRNA.

Fig 6A: Dtprp is still used in the axis lettering without any explanation – change to Prl8a2.

Round 3

Reviewer 1 Report

All necessary corrections and improvements have been properly included in the present manuscript.

Congratulations to the authors for this thorough study!